# Precipitation Kinetics of Water-Cooled Copper Mold Al-Mg-Si(-Mn, Zr) Alloy during Aging

**DOI:** 10.3390/ma16237424

**Published:** 2023-11-29

**Authors:** Hua Shen, Jianchao Shi, Yukun Zhou, Xiaofeng Wang, Guangchun Yao

**Affiliations:** 1School of Science, Shenyang University of Chemical Technology, Shenyang 110142, China; z19951021@126.com (Y.Z.); wind850820@163.com (X.W.); 2Guizhou Aerospace Tianma Electromechanical & Technology Co., Ltd., Zunyi 563000, China; wymy1983@126.com; 3School of Metallurgy, Northeastern University, Shenyang 110819, China; gcyao@mail.neu.edu.cn

**Keywords:** Al-Mg-Si(-Mn,Zr) alloy, water-cooled copper mold, age hardening curve, mechanical properties, activation energy, kinetic equation

## Abstract

The aging precipitation behavior of 6061 aluminum alloy that underwent iron casting and water-cooled copper casting and 6061 aluminum with Mn and Zr elements added was studied. Firstly, the hardness curves, tensile properties, and fracture morphology of four aging alloys—6061 (iron mold casting), 6061 (water-cooled copper mold casting), 6061-0.15Mn-0.05Zr (iron mold casting), and 6061-0.15Mn-0.05Zr (water-cooled copper mold casting)—were studied. The results of the aging hardness curve show that the aging precipitated phase of the 6061 alloy cast with a water-cooled copper mold is dispersed. The addition of Mn increases the amount of coarse inclusion α-(AlMnFeSi) in the alloy, resulting in a decrease in the age hardening property. The addition of Zr is related to the nucleation and growth of the G.P. region in the early aging period, mainly changing the formation rate and quantity of the G.P. region, leading to the advancement of peak aging and an increase in hardness. After the G.P. region gradually transforms into the β phase, the hardness of the alloy increases with the increase in the volume fraction of the β phase. When the β″ phase is coarsened to the point where the fault line can be bypassed, the transitional metastable β′ phase begins to precipitate, and the coherent distortion around it weakens, indicating over-aging. Finally, the equilibrium phase Mg_2_Si is formed. The results of the tensile tests indicate that the tensile strength and yield strength of the 6061-0.15Mn-0.05Zr alloy produced by water-cooled copper casting after aging are 356 Mpa and 230 Mpa, respectively. These values are 80 MPa and 75 MPa higher, respectively, than those of the 6061 aluminum alloy produced via iron casting. However, the elongation is by 5%. The fracture morphology of the tensile sample of the aging alloy shows that dislocation slip in the alloy results in dislocation plugging, stress concentration, and the initiation of crack cleavage on the surface. The fracture of the water-cooled copper mold-casting alloy is a ductile fracture of the microporous aggregation type, and the macroscopic fracture exhibits an obvious “neck shrinkage” phenomenon. The fracture analysis is consistent with the mechanical properties. The DSC curve shows that there is no enrichment process of solute atoms during the heating process, and the aging precipitation process after homogenization is as follows: G.P. zone → β″ phase → β′ phase. The aging precipitation process of the water-cooled copper casting alloy after homogenization treatment is as follows: β″ phase → β′ phase (no precipitation in the G.P. zone was observed). The results of the differential scanning calorimetry (DSC) analysis show that the main strengthening phase in the experimental alloy system is the β″ phase. The activation energies for the β″ phase precipitation were calculated and found to be 147 KJ/mol, 217 KJ/mol, 185 KJ/mol, and 235 KJ/mol, respectively. Additionally, a kinetic equation for the β″ phase precipitation during alloy aging was fitted.

## 1. Introduction

Al-Mg-Si(6XXX) aluminum alloy is one of the most important alloys, and it is a typical heat-treatable alloy. Its strengthening mechanism and performance improvement have always been hot topics in material research. The strengthening of the alloy is mainly due to the small aging strengthening phase (i.e., the second phase) precipitated during the aging process, which hinders the dislocation movement. Therefore, the morphology, size, quantity, and distribution of the aging strengthening phase have important effects on the strength of the aluminum alloy. The precipitation of the aging strengthening phase is also related to the chemical composition, aging temperature, aging time, and microstructure state of the alloy before aging. Thus, the quantitative study of the influence of process parameters on the aging strengthened phase has significant application value for optimizing the alloy composition and improving the alloy properties [1,2,3,4,5]. The alloy melt obtained via copper water-cooled mold casting (CWMC) undergoes a sub-rapid solidification process, which can refine the grains, reduce the second phase size, and improve the solid solubility of alloy solute elements in the matrix. This results in a supersaturated solid solution and a more uniform structure, improving the order of phase precipitation [4,5,6,7,8]. The concept of sub-rapid solidification has evolved alongside the development of rapid solidification. Traditional solidification theory and technology primarily focus on the casting process of ingots and castings, with cooling rates typically ranging from 10^−3^ to 10 K/s. This process is characterized by non-equilibrium solidification, often resulting in the formation of metastable phases. The grain size in the material is closely linked to the cooling rate, where higher cooling rates lead to finer grains and a smaller distance between secondary dendrite arms. This finer microstructure in the cast billet offers a more effective grain refinement effect compared to the conventional alloying method. Furthermore, the sub-rapid solidification method not only allows for a highly uniform chemical composition with increased solid solubility of alloying elements but also inhibits second-phase precipitation and promotes the formation of a non-equilibrium structure. This significantly enhances the strength and plasticity of the alloy, mitigating the harmful and unpredictable micro-battery phenomena associated with conventional alloying. Additionally, it improves the corrosion resistance of the alloy [9]. In this paper, we opted for the traditional solidification method, using an iron mold, in contrast to the sub-rapid solidification method, which employs a water-cooled copper mold.

The study of precipitation in the Al-Mg-Si alloy during aging has been ongoing for approximately 70 years. However, due to its complex precipitation process, the type and sequence of the precipitated phase are influenced by the alloy composition, aging conditions, and processing. Particularly, changes in metastable precipitates directly impact the strengthening effect of materials. Therefore, the aging precipitation behavior of Al-Mg-Si alloy remains a focal point in alloy research. The atomic numbers of Al, Mg, and Si in Al-Mg-Si alloys are close, making it impossible to obtain precipitation information by using conventional X-ray or electron diffraction methods. Additionally, accurately determining the crystal structure of the rod-shaped precipitated β′ phase in the alloy poses a challenge [10].

For the equilibrium Al-Mg-Si alloy (Mg/Si atomic ratio of 2), the aging precipitation sequence is as follows [11,12]:α supersaturated solid solution (SSSS) → Guinier–Preston (G.P.) region → β″ needle→β′ rod → β sheet (equilibrium phase Mg_2_Si).

At the onset of aging, Mg and Si atoms aggregate to form a solute atom-rich region known as the G.P. zone. The G.P. zone consists of spherical Mg, Si, or Mg-Si atoms with no independent lattice structure [13]. Due to the small size of these atoms or groups of atoms, the aggregation process cannot be observed. The formed G.P. zone maintains a coherent relationship with the aluminum matrix, and atoms at the boundary belong to both the parent phase and the G.P. zone. To accommodate these arrangements, elastic strain develops near the coherent boundary, causing lattice distortion and hindering dislocation movement. Consequently, the hardness of the aluminum alloy increases upon G.P. zone formation [11]. As the aging process continues, solute groups tend to organize and grow rapidly, forming fine needle-like β′′ phases, which remain coherently embedded in the matrix. When β′′ reaches a certain size, its stress field extends throughout the matrix, and the alloy hardness peaks. Subsequently, with further enrichment of Mg and Si atoms, the β′′ phase gradually loses its alloy coherent relationship with the matrix. The formed β′′ phase locally retains coherence with the matrix, leading to reduced elastic strain in the surrounding matrix, diminished dislocation obstacles, and a decrease in alloy hardness. In the later stage of aging, the β′ phase and the aluminum matrix lose their coherent relationship, eventually forming a stable equilibrium β (Mg_2_Si). The β phase becomes completely detached from the matrix, coherent strain disappears, and the hardness decreases. Among all strengthening phases, the β′′ phase exhibits the most potent strengthening effect, followed by the β′ phase and β phase [11].

If the Al-Mg-Si alloy contains excessive Si (Mg/Si atomic ratio < 2), the precipitation sequence is as follows [14]:α supersaturated solid solution (SSSS) → G.P. region→β″ needle → β′ + Q′ (strip) → β + Si.If Cu is present in the excess Si alloy, the precipitation sequence is as follows [15]:α supersaturated solid solution (SSSS) → G.P. region → β′ → Q′ → Q (strip) + Si.Or [14,15,16,17]:α supersaturated solid solution (SSSS) → G.P. region (Cu/Al rich) → θ″ → θ′ → θ.

In fact, the aging strengthening of Al-Mg-Si alloy is a very complicated process. Its aging precipitates are composed of one or several β, Q, and θ equal metastable phases, and the aging precipitation process cannot be accurately described by a single precipitation mode.

Differential scanning calorimetry (DSC) is a powerful technique for studying phase transitions and is widely used to correlate the microstructure and properties of Al-Mg-Si alloys with the aging process [18,19,20,21,22,23]. Age hardening in these alloys occurs as a result of metastable precursors of the equilibrium phase β-Mg_2_Si precipitating in a specific order, which can be identified by the study of peak arrangement in the DSC curve [24]. Since Al-Mg-Si(6XXX) series alloys contain a variety of alloying elements, each of which plays a different role in the alloy, metastable precipitates are precipitated in one or more sequences [11,25,26,27]. Due to the complexity of metastable phase precipitation process, the specific precipitation process is not yet fully understood. Therefore, studying the precipitation sequence type, metastable precipitation species, and structural composition of the aging process is of great significance [28,29,30]. Studies have shown that [31,32,33], the change in different treatment states significantly impacts the precipitation process of metastable precipitates formed during subsequent artificial aging. Although the mechanism of artificial aging in the Al-Mg-Si(6XXX) alloy has been reported in the relevant literature [11,24,31,34] and has laid a certain theoretical foundation for exploring the aging precipitation behavior of the Al-Mg-Si alloy, the existing research results are not perfect due to the complexity of the aging precipitation process.

The work presented in this paper is a continuation of previous research. The preliminary work involved selecting the 6061 alloy and 6061 alloy with Mn and Zr as the research object. Four test alloys were designed: (a) iron mold casting 6061 alloy, (b) water-cooled copper mold casting 6061 alloy, (c) iron mold casting 6061-0.15Mn-0.05Zr alloy, and (d) water-cooled copper casting 6061-0.15Mn-0.05Zr alloy. The effects of Mn, Zr, and water-cooled copper casting on the microstructure and mechanical properties of the alloy were analyzed by studying the original microstructure and solid solubility. Water-cooled copper casting was found to reduce element segregation and improve the solid solubility of main alloy elements. For example, the solid solubility of Mg increased by 7–12%, and the solid solubility of Si increased by 10–14%. Additionally, Mn’s solid solubility increased by 10%. The crystal phases of the original alloy were determined through an EDS analysis: the water-cooled copper mold casting process resulted in granular-phase α-Al_8_(MnFe)_2_Si and α-Al_8_(MnFeCr)_2_Si, and strip phase α-Al_9_(MnFe)_3_Si_2_. Common iron mold casting alloys exhibited skeletal phases, Al_5_(MnFe)Si and Al_5_(MnFeCr)Si; granular phases, α-Al_8_(MnFe)_2_Si and α-Al_8_(MnFeCr)_2_Si; and a strip phase, α-Al_9_(MnFe)_3_Si_2_. To eliminate the inhomogeneity and non-equilibrium structure, the as-cast alloy underwent homogenization through heat treatment at 560 °C × 3 h. After homogenization, the morphology of precipitated phases changed to strip-phase β-Al_9_Fe_2_Si_2_ and granular-phase α-Al_8_(FeMnCr)_2_Si, leading to dispersed crystal phases and improved mechanical properties. The influence of homogenization treatment and water-cooled copper mold casting on the mechanical properties of the alloy was analyzed through tensile tests. The results indicated that the homogenized alloy exhibited better mechanical properties than the cast alloy and the water-cooled copper mold casting 6061-0.15Mn-0.05Zr alloy, with a tensile strength of 286 Mpa, yield strength of 127 Mpa, and elongation of 17.84% [35]. In this study, the four alloys after homogenization were immediately artificially aged. The results revealed that the tensile strength, yield strength, and elongation of the 6061-0.15Mn-0.05Zr alloy cast using a water-cooled copper mold after aging were 356MPa, 230MPa, and 21%, respectively. The aging method significantly improved the mechanical properties of the alloy and greatly enhanced the strength of the 6061 alloy. Subsequently, the apparent activation energy of the metastable phase was calculated by differential scanning calorimetry (DSC), and the aging kinetics equation was fitted. The aging precipitation mechanism of the four alloys was discussed. The main strengthening phase of the experimental alloy system is β″ phase, and the precipitation activation energy of β″ phase is calculated in this paper. For 6061 (iron mold casting), 6061 (water-cooled copper mold casting), 6061-0.15Mn-0.05Zr (iron mold casting), and 6061-0.15Mn-0.05Zr (water-cooled copper mold casting) alloys, the apparent activation energies of the β″ phase are 147 KJ/mol, 217 KJ/mol, 185 KJ/mol, and 235 KJ/mol, respectively. Finally, the kinetic equation of the β″ phase precipitation during alloy aging was fitted, providing a theoretical basis for further study of the aging precipitation behavior of the 6061 alloy. In this paper, four types of alloys were artificially aged immediately after homogenization. The microstructure of these four aluminum alloys was observed, and the tensile tests were carried out. Additionally, using differential scanning calorimetry (DSC), the main strengthening phase of the experimental alloy system was determined to be the β″ phase. The activation energy of the β″ phase precipitation was calculated, and a kinetic equation for the β″ phase precipitation during the aging process was fitted. Simultaneously, the aging precipitation mechanism of the four alloys was discussed. This study provides a theoretical basis for further research on the aging precipitation behavior of the 6061 alloy. The innovation of this experiment lies in the use of a self-designed water-cooled copper mold die to enhance the speed. In terms of alloying, we selected and added the appropriate amounts of Mn and Zr elements simultaneously to the 6061 alloy. Subsequently, the apparent activation energy of the metastable phase was calculated using differential scanning calorimetry (DSC), and the aging kinetics equation was fitted. The aging precipitation mechanism of the four alloys was discussed.

## 2. Materials and Methods

### 2.1. Materials and Specimens

The 6061 aluminum alloy was placed in a high-purity graphite crucible and melted in a medium-frequency electromagnetic induction vacuum melting furnace. It was heated to 750 °C, and Al-6 wt.%Zr, Al-9 wt.%Mn, and Al-20 wt.%Si intermediate alloys were added sequentially. The mixture was stirred with a graphite rod for 1 min, left to cool until the furnace temperature reached 720 °C, and held at that temperature for 10 min. Hexachloroethane was then added and gently stirred with a graphite rod for 1–2 min. The temperature was raised to 780 °C and maintained for 10 min. After slag removal, the ingot was obtained by casting it into both iron and water-cooled copper molds, respectively. The structure diagram of independently designed water-cooled copper mold castings is shown in Figure 1 (note: the unit is millimeter).

The actual composition of the as-cast alloy was determined using a full-spectrum direct reading inductance coupling Plasma Emission Spectrometer (ICP, Perkin Elmer, Plasma 400, PerkinElmer, Inc., Shanghai, China). The test results are presented in Table 1.

The water-cooled copper mold ingot measures 100 mm × 30 mm × 150 mm, while the iron mold ingot has dimensions of Φ40 mm × 100 mm. The copper ingot was cut along the cross-section, 30 mm from the bottom, to obtain a 10 mm × 10 mm × 10 mm cube and a tensile sample, both taken from the center of the section. The iron mold ingot is cut along the longitudinal section of the center of the circle, and the sample is taken at the center of the section.

### 2.2. Testing Method

After the homogenization of the four alloys, the aging process was carried out in a QH2002A constant-temperature drying oven. The potentiometer was calibrated to maintain the furnace temperature within ±1 °C. The aging process was conducted at 175 °C. Vickers hardness was measured by cooling in water to room temperature. The hardness tests were performed using a 450SVD Vickers hardness tester with a 5 kg load and a load retention time of approximately 10 s. For each hardness test, the test value represents an average of 6 measurements.

The flake samples for as-cast drawing are cut from the bottom of the ingot and processed using electric spark wire cutting. Three samples were taken for each state, and the size and shape of these samples are illustrated in Figure 2 (note: the unit is millimeter). The tests were conducted on the CMT 5105-SANS 100 kN electronic universal test machine in accordance with the GB/T 228-2002 standard [36] (The People’s Republic of China National Standard for the metal tensile test method). The standard distance was set at 25 mm, the strain rate was 6.67 × 10^−4^/s, and the tensile speed was 0.5 mm/s. Three samples were measured for each state, and the results were averaged. Three samples were measured for each condition, and the results were averaged. The stretching port was used for protection, and microstructure observation was carried out after mounting. The fracture morphology of the tensile specimens of the aging alloys was observed by scanning electron microscopy (SEM).

The DSC test sample was mechanically cut to form a disc with a diameter of 4 mm and a thickness of 0.5 mm. The disc was then smoothed with 1200# sandpaper until the surface was clean and bright, rinsed with alcohol, and blotted dry with filter paper. The thermal analysis was conducted using an STA 449C thermal analyzer. Pure aluminum crucibles were used, with a pure aluminum crucible as the reference sample, and nitrogen was used for protection. The heating rate was set to 5 °C/min.

## 3. Results and Discussion

### 3.1. Age Hardening Curve of Al-Mg-Si(-Mn,Zr) Alloy

Figure 3 shows the hardening curves of 6061 aluminum alloy and 6061-0.15Mn-0.05Zr aluminum alloy, both of which were homogenized using iron mold casting and water-cooled copper mold casting at 175 °C. In Figure 3a, there are two incubation periods of slow changes in hardness value before its peak: one at 15 min and another 50 min. During the initial 20 min of aging, the alloy’s hardness increased by 4.5 HV. After aging for 30 min, the hardness reaches its lowest point before it begins to rise again, ultimately reaching its peak hardness at 720 min. The alloy’s maximum hardness value is 114 HV. In Figure 3b, the incubation period is approximately 50 min, after which the hardness value increases over the course of 660 min. At peak aging, the alloy reaches a maximum hardness of 119 HV. For water-cooled copper cast 6061 alloy, the aging-precipitated phase is distributed and dispersed, while iron casting experiences significant segregation, resulting in higher hardness. Figure 3c depicts a decreasing trend in hardness at the beginning of aging, possibly attributed to the dissolution of smaller G.P. regions in the structure, leading to a reduction in hardness. The alloy exhibits an incubation period of approximately 15 min, followed by reaching peak aging at 540 min, with a maximum hardness value of 116 HV. Compared to Figure 3a, the addition of Mn and Zr elements results in an increase in the hardness of the homogenized alloy by approximately 3 HV, while the aging curve decreases. However, the maximum hardness difference is only 2 HV, which is essentially the same. The reason for this lies in the fact that the addition of Mn increases the presence of coarse inclusions, specifically α-(AlMnFeSi), within the alloy, resulting in a reduction in the age-hardening properties [10]. On the other hand, the addition of Zr is associated with the nucleation and growth of G.P. regions during the early aging stage. This primarily affects the formation rate and quantity of G.P. regions, leading to advanced peak aging and an increase in hardness [5,37,38,39]. In Figure 3d, three plateaus are observed at 5–10 min, 40–50 min, and 6–7 h, eventually reaching the maximum peak value of 118 HV at 630 min. With prolonged holding times, the hardness value decreases. The initial increase in hardness is attributed to the formation of the G.P. zone, which subsequently undergoes a gradual transformation into the β″ phase. As the volume fraction of the β″ phase increases, so does the hardness of the alloy [40]. As the aging process continues, when the β″ phase coarsens and fault lines can be bypassed, the transitional metastable β′ phase begins to precipitate. However, it no longer maintains a complete coherent relationship with the parent phase after precipitation. At this stage, the coherent distortion within the alloy weakens, resulting in a reduction in hardness, and the alloy becomes over-aged. Ultimately, the equilibrium phase Mg_2_Si is formed [41].

### 3.2. Tensile Properties and Fracture Morphology of Al-Mg-Si(-Mn,Zr) Aged Alloys at Room Temperature

The homogenized tensile sample was artificially aged (at 175 °C), and the room temperature tensile test was conducted immediately after ageing. Figure 4 presents the room temperature tensile properties of 6061 and 6061-0.15Mn-0.05Zr aluminum alloys aged by both iron casting and water-cooled copper casting. After artificial aging following homogenization, the alloy’s tensile strength and yield strength showed significant improvement. For the 6061 alloy, after aging, the tensile strength and yield strength are 276 MPa and 311 MPa, respectively, for both iron mold casting and water-cooled copper mold casting. The elongation percentages are 26% and 36%, respectively, for these two casting methods. For the 6061-0.15Mn-0.05Zr alloy, after aging, the tensile strength and yield strength are 280 MPa and 356 MPa, respectively, for iron casting; for water-cooled copper casting, they are 152 MPa and 230 MPa, respectively. The elongation percentages are 22% and 21%, respectively. References [42,43] suggest that the strength and toughness of the material are primarily influenced by the interaction between the size and distribution of the β″ precipitated phase and the dislocations. When Mn and Zr elements are added to the 6061 alloy, and for the case of using a water-cooled copper mold, they promote the presence of high dislocation density and the precipitation of a nanosized β″ phase [44,45]. As a result, the tensile strength of the 6061 alloy and the 6061-0.15Mn-0.05Zr alloy, when cast with a water-cooled copper mold, can reach 311 MPa and 356 MPa, respectively, with elongation values of 36% and 21%. This leads to improved strength and good toughness.

Figure 5 shows the fracture morphology of the tensile specimens of an aging alloy. After the aging of the alloy, the fracture of the 6061 alloy in iron mold casting exhibited numerous dimples (Figure 5a, A), and the bottom size was relatively large (Figure 5a, B), possibly caused by the presence of the second phase or inclusions. However, the 6061 alloy cast with a water-cooled copper mold showed a small number of slip bands and “tearing edges” (Figure 5b, A), suggesting potential dislocation slip in the alloy, leading to dislocation plug, resulting in stress concentration and the initiation cracks. In comparison to Figure 3a, the fracture of the 6061-0.15Mn-0.05Zr alloy cast with an iron mold (Figure 5c) displayed a smaller dimple size and a “lamellar ridge” (Figure 5c, A), indicating that the addition of Mn and Zr refined the grain. Simultaneously, Mn has the effect of melting impurity Fe and can form Al_6_(Fe, Mn), reducing the occurrence of the second phase and inclusions. Consequently, the addition of Mn and Zr enhanced the toughness and strength of the alloy, aligning with the tensile properties illustrated in Figure 4. A local cleavage surface also appeared, indicating the initiation of an initial crack. The fracture of the 6061-0.15Mn-0.05Zr alloy cast with a water-cooled copper mold (Figure 5d) demonstrated a microporous aggregation ductile fracture, with small and uniform dimple sizes (Figure 5d, A). This suggested that the grain refinement effect is more pronounced under the combined action of water-cooled sub-rapid solidification and the alloying of Mn and Zr, resulting in the maximum tensile strength. The fracture analysis is consistent with the mechanical properties.

### 3.3. Kinetics of Aging Precipitation of Al-Mg-Si(-Mn,Zr) Alloy

#### 3.3.1. DSC Analysis of Al-Mg-Si(-Mn,Zr) Alloy

The literature [11] points out that, in the case of the Al-Mg-Si-Cu-Mn alloy, there exist a certain number and size of G.P.-region supersaturated solid solutions in the T4 alloy state. Therefore, when DSC is applied, a series of changes will occur in the microstructure of this state. Studies in the literature [46,47,48,49] indicate that the DSC curve of the Al-Mg-Si alloy in the T4 typically consists of four exothermic peaks:Exothermic peak 1: Mg, Si, MG-Si atomic clusters;Exothermic peak 2: G.P. zone;Exothermic peak 3: β″ phase precipitation;Exothermic peak 4: β′ phase precipitation.

Figure 6 shows the DSC curves of 6061 and 6061-0.15Mn-0.05Zr aluminum alloys homogenized via iron casting and water-cooled copper casting at temperatures ranging from 50 °C to 500 °C. In Figure 4, no obvious exothermic peaks appeared in the low-temperature region (50–100 °C), indicating that solute atom enrichment did not occur during the DSC heating process. This is because the atomic clusters formed during the early stages of artificial aging have reached a stable state. At this point, there are no excess solute atoms or vacancies available to form atomic groups within the solid solution, resulting in the absence of exothermic peaks in the low-temperature region (50–100 °C). In Figure 6a, a small endothermic peak appears in the range of 120 to 160 °C, indicating the dissolution of the G.P. region during the artificial aging process. The wide heat absorption peak is attributed to the non-uniform size of atomic groups within the G.P. region, which is a result of iron mold casting. The peak at 246 °C corresponds to the precipitation of the acicular β″ phase, while the exothermic peak at 285 °C represents the transition phase between the β″ phase and the β′ phase. Finally, the peak at 331 °C indicates the precipitation of the β′ phase. In Figure 6b, there are no low-temperature precipitation peaks and no G.P. region-formation peaks, meaning that there are no exothermic peaks or absorption peaks before the temperature drops below 200 °C. This suggests that the processes of atomic enrichment and G.P. region formation were completed prior to DSC heating. The reason for this is that water-cooled copper mold casting slows down the rate of atomic diffusion and migration, resulting in fewer solute element precipitations. During the cooling process, atom enrichment and G.P. zone formation occurred rapidly. The first exothermic peak (247 °C) appears in the range of 215 °C to 260 °C, signifying the precipitation of the β″ phase. This indicates that the saturation of solute atoms in the matrix promotes the nucleation of the β″ phase. As the temperature continues to rise, a second exothermic peak appears at around 289 °C, resulting from the transformation of the β″ phase into the β′ phase. In Figure 6c, the small exothermic peak in the range of 165 °C to 200 °C is caused by the formation of the G.P. region. The formation of the G.P. region results from the large number of atomic clusters formed before DSC heating which increase in size during the heating process. As the temperature continues to rise, the G.P. region, which serves as the center of phase nucleation, transforms into the β phase, leading to an exothermic peak at approximately 251 °C. The exothermic peak at 310 °C corresponds to the precipitation of the β′ phase, while the small and broad exothermic peak at 291 °C represents the transition phase during the transformation from the β″ phase to the β′ phase, a transition related to the presence of Mn and Zr elements. The peak shape and temperature of Figure 6d are similar to those of Figure 6b. However, with the addition of Mn and Zr elements, the β″ phase shifts towards higher temperatures, while the β′ phase shifts toward lower temperatures. In summary, the addition of Mn and Zr elements weakens the nucleation driving force of β″ and increases the activation energy, causing the β″ phase to lag and shift towards higher temperatures. The introduction of Mn and Zr elements promotes the precipitation of the β′ phase, leading to a slight advancement in β′ phase precipitation and, consequently, an earlier exothermic peak. The sub-rapid solidification induced by water cooling indicates agglomeration in the G.P. region, which is more conducive to nucleation.

#### 3.3.2. Activation Energy and Kinetic Equation for Al-Mg-Si(-Mn,Zr) Alloy

The DSC results show that β″ phase is the primary strengthening phase of the experimental alloy. The activation energy, *Q**, of the β″ phase in the experimental alloy is calculated, and, subsequently, the kinetic equation is derived.

For isothermal transformation processes, the kinetics of phase transformation can be described using the Johnson–Mehl–Avrami (J-M-A) Equation (1) [11,46,50,51]:(1)$$\nu =1-\exp(-kt^{n})$$
where *v* is the volume fraction of precipitates newly increased in time, *t*; *k* is a temperature-dependent parameter; and *n* is the nucleation growth coefficient.

For non-isothermal transition processes, Formula (1) is modified to Formula (2):(2)$$\nu =1-\exp(-k_{1}^{n}t^{n})$$

Then, *k*_1_ follows the Arrhenius formula, Formula (3):(3)$$k_{1}=k_{0}\exp(-Q^{*}/RT)$$
where *k*_0_ is a constant; *Q** is activation energy, measured in J/mol; R is the gas constant, with a value of 8.314 J/(K·mol); and T is the thermodynamic temperature, expressed in Kelvin (K).

Calculating the time derivative of Formula (2) yields Formula (4) for the volume fraction transformation rate in the non-isothermal process:(4)$$\frac{d\nu }{dt}=k_{1}(T)V(v)$$
where *V*(v) is an implicit function of v, and by combining it with Formula (3) and Formula (4), we can obtain the expression for the implicit function (5):(5)$$V(v)=n(1-v){\left[-\ln\frac{1}{1-v}\right]}^{\frac{n-1}{n}}$$

The literature [46] shows that, when the precipitates are needle-like or strip-like, n = 1, and Formula (5) is simplified to Formula (6):(6)V(v)=1−v

At temperature *T*, the integral value of the precipitated object can be obtained according to DSC experimental results, that is, Formula (7):(7)$$v(T)=\frac{A(T)}{A(T_{f})}$$
where *A*(*T*) is the area of baseline and the DSC curve peak at temperature *T*, while *A*(*T_f_*) is the entire peak area.

Assuming that the thermal analytic formula of the DSC curve peak is represented by *y*(*T*), the peak area at temperature *T* is as shown in Formula (8):(8)$$A(T)=\int_{Ti}^{T}y(T)dT$$

Subsequently, we can derive the relationship between the volume transition, *v*, and temperature, *T*, represented as Formula (9):(9)$$v(T)=\frac{\int_{T_{i}}^{T}y(T)dT}{\int_{T_{i}}^{T_{f}}y(T)dT}=\frac{\int_{T_{i}}^{T}y(T)dT}{A(T_{f})}$$

Then, we take the derivative of Formula (9) to obtain Formula (10):(10)$$\frac{dv}{dT}=\frac{y(T)}{\int_{T_{i}}^{T_{f}}y(T)dT}$$

If the heating rate, $\frac{dT}{dt}$, is constant and expressed by *β*, the relationship between the volume transformation, v, and time, *t*, is as shown in Formula (11):(11)$$\frac{dv}{dt}=\frac{dv}{dT}\frac{dT}{dt}=\frac{y(T)}{\int_{T_{i}}^{T_{f}}y(T)dT}•\beta =\frac{y(T)}{A(T_{f})}•\beta$$

The heating rate is *β* = 5 K/min, and Formula (12) is obtained from Formula (3), Formula (4), and Formula (11):(12)$$\ln[\frac{y(T)}{\int_{T_{i}}^{T_{f}}y(T)}•\frac{1}{12}]•\frac{1}{1-\frac{\int_{T_{i}}^{T}y(T)d(t)}{\int_{T_{i}}^{T_{f}}y(T)dT}}=\ln k_{0}-\frac{Q^{*}}{RT}$$

The quadratic function *y(T)* = a*T*^2^ + b*T* + c was chosen to fit the DSC curve. Within the temperature range where the peak occurs, we select various temperatures, *T*, and insert them into Formula (12) to obtain a series of values. Subsequently, using 1/*T* as the horizontal coordinate and the corresponding value from the formula as the vertical coordinate, we then create a straight line. The transition activation energy of the precipitated phase is determined from the slope of the line, and the constant, *k*_0_, is obtained from the intercept.

The preceding results indicate that the β″ phase is the primary metastable phase during the aging process of the alloy in this experiment (refer to Figure 4). Consequently, the activation energy obtained for the precipitation of the β″ phase is presented in Table 2.

The literature [11,52] points out that the precipitation of the β″ phase promotes the diffusion of Mg and Si atoms through supersaturated vacancies. The activation energy of the water-cooled copper mold casting alloy is higher than that of the iron mold casting alloy, indicating that the β″ phase is challenging to precipitate. This is because alloying elements cannot be fully diffuse during water-cooled copper mold casting, resulting in a reduced number of precipitated phases that are difficult to form. This observation aligns with the microstructure analysis.

Table 3 presents the relevant data of β″ precipitated phase on the DSC curve of the a ~d alloys after homogenization.

Therefore, the kinetic equation describing the behavior of the four alloys is as follows:(a)$\frac{dv}{dt}=-0.21+1.70\times 10^{-3}t-0.35\times 10^{-5}t^{2}$(b)$\frac{dv}{dt}=-0.64+5.15\times 10^{-3}t-1.06\times 10^{-5}t^{2}$(c)$\frac{dv}{dt}=-0.39+3.14\times 10^{-3}t-0.63\times 10^{-5}t^{2}$(d)$\frac{dv}{dt}=-0.70+5.67\times 10^{-3}t-1.13\times 10^{-5}t^{2}$

## 4. Conclusions

i.At an aging temperature of 175 °C, the peak aging times for 6061 aluminum alloy and 6061-0.15Mn-0.05Zr aluminum alloy, cast using iron and a water-cooled copper mold, are 720 min, 660 min, 540 min, and 630 min, respectively. The aging process significantly improves the tensile strength and yield strength of the alloy. Specifically, for the 6061-0.15Mn-0.05Zr alloy cast using a water-cooled copper mold after aging, the tensile strength yield strength is 356 MPa, the yield strength is 230 MPa, and the elongation is 21%.ii.The DSC curves of the four experimental alloys did not exhibit clear absorption peaks in the low-temperature region (50–100 °C), thus indicating the absence of a solute atom enrichment process during the DSC temperature rise. The aging precipitation process for the iron mold casting alloy after homogenization can be summarized as follows: GP zone → β″ phase → β′ phase. In contrast, the aging precipitation process for the water-cooled copper casting alloy after homogenization treatment proceeds as follows: β″ phase → β′ phase (no precipitation in the GP region was observed).iii.The activation energies for the four experimental alloys were calculated, and their respective kinetic equations were determined. The precipitation activation energies of the β phase in these alloys are as follows: 147 KJ/mol, 217 KJ/mol, 185 KJ/mol, and 235 KJ/mol. These values further validate the aging precipitation process of water-cooled copper mold casting alloys from a thermodynamic perspective.

## Figures and Tables

**Figure 1 materials-16-07424-f001:**
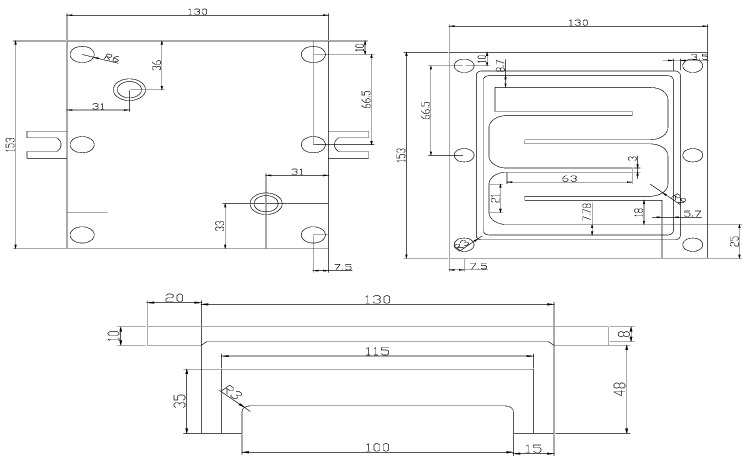
Diagrammatic sketch of water-cooled Cu mold.

**Figure 2 materials-16-07424-f002:**
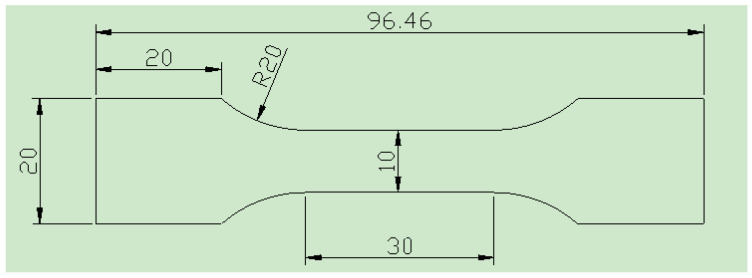
The shape and size of plate tensile samples.

**Figure 3 materials-16-07424-f003:**
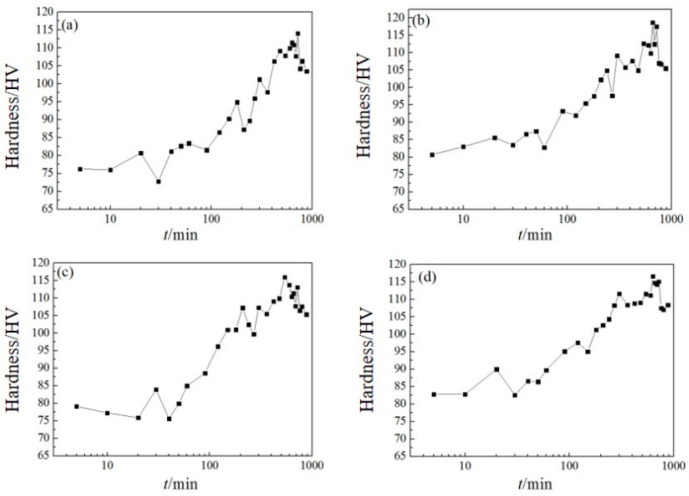
Aging curves of cast-T4 specimens aged at 175 °C: (**a**) 6061(Fe), (**b**) 6061(Cu), (**c**) 6061-0.15Mn-0.05Zr(Fe), and (**d**) 6061-0.15Mn-0.05Zr(Cu).

**Figure 4 materials-16-07424-f004:**
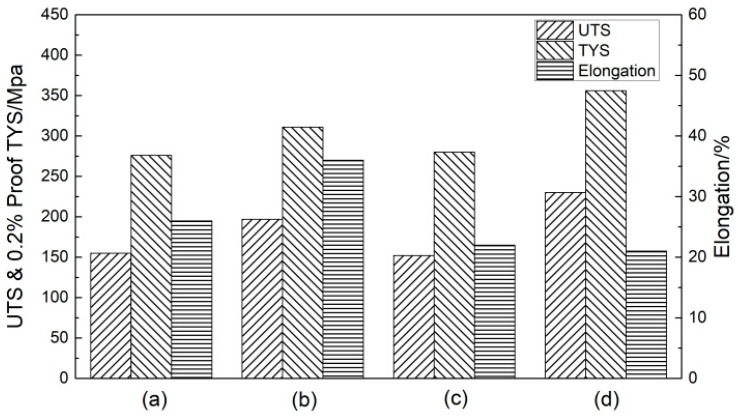
Tensile properties of aged 6061 alloys and 6061-0.15Mn-0.05Zr alloys: (**a**) 6061(Fe), (**b**) 6061(Cu), (**c**) 6061-0.15Mn-0.05Zr(Fe), and (**d**) 6061-0.15Mn-0.05Zr(Cu).

**Figure 5 materials-16-07424-f005:**
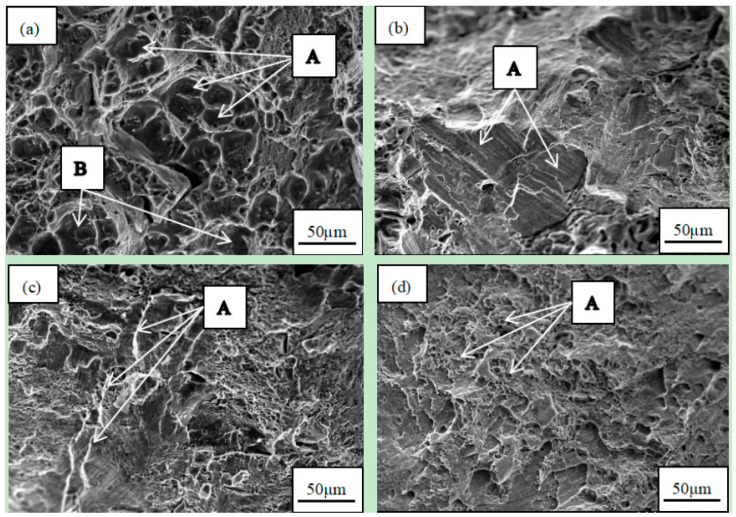
SEM images of fractographs for aged 6061 and 6061-0.15Mn-0.05Zr alloys: (**a**) 6061(Fe), (**b**) 6061(Cu), (**c**) 6061-0.15Mn-0.05Zr(Fe), and (**d**) 6061-0.15Mn-0.05Zr(Cu).

**Figure 6 materials-16-07424-f006:**
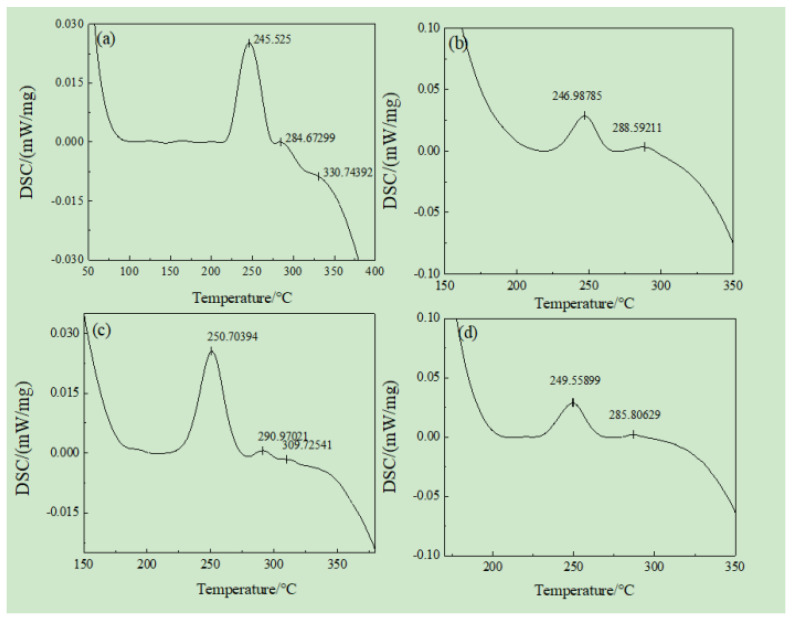
DSC curves of homogenization-treated 6061 and 6061-0.15Mn-0.05Zr alloys: (**a**) 6061(Fe), (**b**) 6061(Cu), (**c**) 6061-0.15Mn-0.05Zr(Fe), and (**d**) 6061-0.15Mn-0.05Zr(Cu).

**Table 1 materials-16-07424-t001:** Chemical composition of experimental alloys (ωt/%).

Alloy (No.)	Mg	Si	Mn	Cu	Ti	Cr	Zn	Fe	Zr	Al
6061, cast with iron mold (a)	0.901	0.737	0.010	0.240	0.014	0.128	0.008	0.192	-	Bal.
6061, cast with copper mold (b)	0.901	0.737	0.010	0.240	0.014	0.128	0.008	0.192	-	Bal.
6061-0.15Mn-0.05Zr, cast with iron mold (c)	0.750	0.615	0.147	0.170	0.013	0.110	0.009	0.320	0.043	Bal.
6061-0.15Mn-0.05Zr, cast with copper mold (d)	0.750	0.615	0.147	0.170	0.013	0.110	0.009	0.320	0.043	Bal.

**Table 2 materials-16-07424-t002:** The activation energy of the β″ phase of the homogenized alloy from a to d.

No.	Alloys	β″, Peak Center Temperature (°C)	Precipitation Activation Energy (kJ/mol)
A	6061(Fe)	246	147
B	6061(Cu)	247	217
C	6061-0.15Mn-0.05Zr(Fe)	251	185
D	6061-0.15Mn-0.05Zr(Cu)	249	235

**Table 3 materials-16-07424-t003:** The correlative data of the β″ precipitate phase on the DSC curve of homogenized alloy from a to d.

No.	Alloys	*T*/°C	1/*T* (1/K)	ln[(d*f*/d*T*)(d*T*/d*t*)/*V*(*v*)]
a	6061(Fe)	230	0.001988072	−6.367079786
		235	0.001968504	−5.980911925
		240	0.001949318	−5.66956535
		245	0.001930502	−5.381901365
b	6061(Cu)	230	0.001988072	−6.517846433
		235	0.001968504	−5.841209366
		240	0.001949318	−5.393880172
		245	0.001930502	−4.992135248
c	6061-0.15Mn-0.05Zr(Fe)	235	0.001968504	−6.465682158
		240	0.001949318	−5.918888148
		245	0.001930502	−5.533390031
		250	0.001912046	−5.194921128
d	6061-0.15Mn-0.05Zr(Cu)	235	0.001968504	−6.470727477
		240	0.001949318	−5.734467332
		245	0.001930502	−5.268079103
		250	0.001912046	−4.855916638

## Data Availability

No new data were created or analyzed in this study. Data sharing is not applicable to this article.

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
