# Peer review of "Precipitation Kinetics of Water-Cooled Copper Mold Al-Mg-Si(-Mn, Zr) Alloy during Aging"

_materials, 2023, doi:10.3390/ma16237424_

Round 1

Reviewer 1 Report

Comments and Suggestions for Authors

Dear Authors

I have read the manuscript titled " Precipitation Kinetics of Water-Cooled Copper Mold Al-Mg-Si(-Mn, Zr) Alloy During Aging " for possible publication in the journal Materials. Following are my comments

1. The abstract needs to be modified. it should bring the summary of the work and datas quantitatively

2. Please rephrase the keywords

3. The introduction needs to be modified. It should contain the novelty of the work to be conducted. Why the mould was selected? The reason behind the selection? This mold differs from others, like sand, metal, and carbon dioxide. All these things should be addressed in the introduction. Moreover, I suggest the authors add the following articles, which are relevant to the field of Al-Si-Mg alloy. Some of the papers mentioned below are very relevant to the sentences that the authors have mentioned in the introduction. Kindly review the articles below, and you may cite them accordingly. Moreover, the introduction lacks details pertinent to the previous study. I request the authors to review the intro part and make it more interesting for the readers. Novelty should be clearly mentioned in the last part of the intro.

10.1007/s12540-021-01054-y 10.1007/s12633-023-02342-5

Experimental methodology

1. How was the composition tested for Table 1? could you please mention the equipment used

2. What is the purpose of choosing 6061?

3. Image of the cast samples will add novelty to the work. Please provide

4. Did the authors add any grain refiner? like Sr or Na? if not why?

Testing methods

1. there is no separate heading for heat treatment. kindly provide

2. " Vickers hardness was measured by cooling in water to room temperature." - what do you mean by this?

3. Heat treatment is not clear at all. Please rewrite. I suggest the authors to rewrite the testing method entirely.

4. details for conducting tensile specimen is also not clear

Results

1. Kindly improve the quality of Fig 1

2. There is no microstructure to substantiate Fig. 1. Kindly provide the same

3. There is no characterization in this article. Kindly use characterisation techniques to substantiate the phases, crystal structure, etc. Please conduct at least XRD, SEM / EDS, 

4. The authors have written the redials and discuss part very vague. no proofs are given to prove them

5. Very poor Fig.2. use Origin to plot the figures. Moreover, discussion on the improve in properties is not sifficient

6. Figure 3 - Mar the features in the SEM image. Discussion needs to be improved

7. Initially, the authors investigated Fe and Cu mold, but why is the title only meant for Cu mold?

8. DSC discussion needs to be improved

9. the authors have generally done good work but failed to analyze the morphology or characterize the samples. Moreover, a lack of detailed discussion is evident in the manuscript. Henceforth, I am recommending MAJOR REVISION for this article. I request authors to address these comments, which will improve the standard off the paper

Comments on the Quality of English Language

extensive English editing required

Reviewer 2 Report

Comments and Suggestions for Authors

The overall quality of this paper is good and the information is of use.  There are two issues.  The first is that there are issues with the english.  For example, The first sentence is awkward: The aging precipitation behavior of 6061 aluminum alloy was studied through iron casting 11 and water-cooled copper processes, along with the inclusion of Mn and Zr elements.  This should say something more like: Differences in the precipitation  behavior of the age-hardening 6061 cast in either steel or water-cooled copper molds were studied.  Process variable also included the intentional inclusion of Mn and Zr as additional alloying elements. 

There are similar needed corrections elsewhere.

The above sentence is connected to the other issue, which is that important experimental details are missing or inadequately described.   Firstly, I suspect the mold referred to as "iron" is most likely "steel.  The particular alloy and its thermal properties should be included.  Also, it appears that the copper mold was a rectangular prism whereas the iron-based cylindrical.  Only the internal dimensions are given.  No cooling curve information is provided.  Yet the nature of the diffusion distance and the ability of the atoms to rearrange during solidification are discussed in some detail.  I feel the authors need to describe their methods more completely for their results to be useable by the community.  If they do not have or cannot obtain cooling data, they should describe analogies to engineering part sizes, provide quantitative estimates, and justify using different geometries with the different mold materials, etc.

Comments on the Quality of English Language

See above

Reviewer 3 Report

Comments and Suggestions for Authors

The paper describes casting of 6061 aluminum alloy with different cooling procedures enhanced with admixture of trace elements Mn and Zr. Casting in a water-cooled mold and subsequent aging at 175C for 9-12h produced an alloy with an increased hardness and tensile strength, which is consequence of needle-like crystals in the alloy β’’ phase. Meaningful results are obtained by differential scanning calorimetry. The authors determined the activation energies for β’’phase precipitation and proposed simple analytical approximation for its kinetics.

The paper is well composed and written in clear understandable language, and it reports important results. Only a few minor flaws were detected:

Lines 33 and 36: aging strengthened phase – better ‘aging strengthening phase’ 

Line 53: G.P. regions – At first mentioning, you may spell the full name ‘Guinier-Preston (G.P.) region’.

Lines 97, 98: Although the mechanism of artificial aging. – need be removed 

Line 162: Water-cooled copper – probably ‘For water-cooled copper’

Line 206: cas using – probably ‘for the case of using’ 

Line 270: authors, please check β or β’’.

Line 308: Eq. (10) seems to be missing. 

Line 309: expressed by β – Due to excessive use, I suggest replacing β with some other symbol in this role.

Line 314: input them – better ‘insert them’

Line 330-331: curve of the a ~ d – do you mean ‘aged’? 

Line 334-338: specify the unit for t (probably minutes).

Round 2

Reviewer 1 Report

Comments and Suggestions for Authors

Dear Authors

Thanks for the revised manuscript. Introduction I had asked you to cite few papers which you haven't done it. Please cite accordingly

10.1007/s12540-021-01054-y

10.1007/s12633-023-02342-5

Fig.1 is not at all clear. Could you provide a much clear image?

In table 1 write wt%

I would like to see the images of the cast rods. Did you detect any blowholes? How was the cast specimen after microanalysis?

In Fig 2. what is the dimension of the specimens? Is it in mm? You should mention that. Moreover, fig is not clear.

If this work is a continuation, kindly discuss that paper in the introduction briefly

Kindly mark the features in Fig 5
